# Study on the Compatibility of Gas Adsorbents Used in a New Insulating Gas Mixture C$_4$F$_7$N/CO$_2$

**Qingdan Huang, Yong Wang, Jing Liu \*, Yaru Zhang and Lian Zeng**

Electric Power Test and Research Institute, Guangzhou Power Supply Co. Ltd., Guangzhou 510410, China; cloveryours@hotmail.com (Q.H.); wangy@guangzhou.csg.cn (Y.W.); zhangyaru1989@163.com (Y.Z.); whu282070193@live.com (L.Z.)

**\*** Correspondence: greengasguangzhou@163.com; Tel.: +86-20-87125506

**Abstract:** An environment-friendly insulating gas, perfluoroisobutyronitrile (C$_4$F$_7$N), has been developed recent years. Due to its relatively high liquefaction temperature (around −4.7 °C), buffer gases, such as CO$_2$ and N$_2$, are usually mixed with C$_4$F$_7$N to increase the pressure of the filled insulating medium. During these processes, the insulating gases may be contaminated with micro-water, and the mixture of H$_2$O with C$_4$F$_7$N could produce HF under breakdown voltage condition, which is harmful to the gas insulated electricity transfer equipment. Therefore, removal of H$_2$O and HF in situ from the gas insulated electricity transfer equipment is significant to its operation security. The adsorbents with the ability to remove H$_2$O but without obvious C$_4$F$_7$N/CO$_2$ adsorption capacity are essential to be used in this system. In this work, a series of industrial adsorbents and desiccants were tested for their compatibility with C$_4$F$_7$N/CO$_2$. Pulse adsorption tests were conducted to evaluate the adsorption performance of these adsorbents and desiccants on C$_4$F$_7$N and CO$_2$. The 5A molecular sieve showed high adsorption of C$_4$F$_7$N (22.82 mL/g) and CO$_2$ (43.86 mL/g); F-03 did not show adsorption capacity with C$_4$F$_7$N, however, it adsorbed CO$_2$ (26.2 mL/g) clearly. Some other HF adsorbents, including NaF, CaF$_2$, MgF$_2$, Al(OH)$_3$, and some desiccants including CaCl$_2$, Na$_2$SO$_4$, MgSO$_4$ were tested for their compatibility with C$_4$F$_7$N and CO$_2$, and they showed negligible adsorption capacity on C$_4$F$_7$N and CO$_2$. The results suggested that these adsorbents used in the gas insulated electricity transfer equipment filled with SF$_6$ (mainly 5A and F-03 molecular sieves) are not suitable anymore. The results of this work suggest that it is a good strategy to use a mixture of desiccants and HF adsorbents as new adsorbents in the equipment filled with C$_4$F$_7$N/CO$_2$.

**Keywords:** perfluoroisobutyronitrile; adsorbents; desiccants; HF removal; insulating gas

## 1. Introduction

Currently, SF$_6$ is the most widely used insulating gas in gas insulated electricity transfer equipment, such as gas insulated switchgear (GIS) and gas insulated line (GIL); however, due to its environmental issues, a new environmentally friendly insulating gas is urgently needed. Perfluoroisobutyronitrile (C$_4$F$_7$N) has been developed as a promising new insulating gas, which shows two times the dielectric strength compared with that of SF$_6$ at the same pressure [1]. Its global warming potential (GWP$_{100}$, 2210 for C$_4$F$_7$N) is clearly lower than that of SF$_6$ (GWP$_{100}$, 23,500), and its atmospheric lifetime is 35 years, which is much shorter than that of SF$_6$ with an atmospheric lifetime of 3200 years [2]. According to the above characteristics, C$_4$F$_7$N could be an alternative gas for SF$_6$ [3]. However, due to its high boiling point (−4.7 °C), buffering gas with low liquefaction temperature, such as N$_2$, CO$_2$, is needed to mix with it for electricity transfer applications [4].

As one of the most widely used insulating gas, SF$_6$ could be decomposed into HF, H$_2$S, SO$_2$, SOF$_2$, etc. with a trace amount of H$_2$O [5–7]. These products are highly toxic, and the acidic gases, such as HF,

$H_2S$ and $SO_2$, are corrosive to the gas insulated equipment, and therefore threaten the security of gas insulated electricity transfer equipment. Many of the regular adsorbents, such as 5A and F-03 molecular sieves, are commonly placed in the $SF_6$ gas insulated electricity transfer equipment to eliminate the moisture, and they are capable of adsorbing acidic gases once produced thermally or by discharge. The research results suggest that $C_4F_7N$ could be thermally decomposed into CO, $COF_2$, $CF_3CN$, $C_2F_5CN$, etc. [8]. The theoretical study results indicate that HF, HCN could be generated by a discharge in the presence of trace $H_2O$ [9]. Therefore, it is significant to control the moisture level of $C_4F_7N$ gas by supplementing desiccants, and it is also beneficial to the security of the equipment to supplement HF adsorbents. As mentioned above, 5A molecular sieve is commonly used as an adsorbent to the decomposed products of $SF_6$, for the reason that it shows good moisture elimination efficiency and acidic gas adsorption capacity [10], and meanwhile, its adsorption capacity of $SF_6$ is quite low. Due to its high boiling point, $C_4F_7N$ needs to mix with $N_2$ and $CO_2$ in application. One should not only evaluate the compatibility of the commonly used adsorbents with $C_4F_7N$, but also evaluate the compatibility of the adsorbents with the buffering gases. As we know, 5A molecular sieve is a good adsorbent for $CO_2$ and $H_2O$ adsorption [10,11], therefore, its compatibility with $C_4F_7N/CO_2$ is suspected and needs to be confirmed. It is also reported that $\gamma$-$Al_2O_3$ is highly effective at adsorbing $C_4F_7N$ [12]. However, the information about adsorbents that could be used for $C_4F_7N/CO_2$ is quite limited.

In this work, in order to study the compatibility of commonly used adsorbents with the new insulating gas $C_4F_7N/CO_2$, a series of adsorbents, including 3A [13], 4A [14,15], 5A [10,16] zeolite molecular sieves, and an adsorbent commonly used in Chinese gas insulated electricity transfer equipment (GIS and GIL), F-03 zeolite molecular sieve, were tested for their adsorption performance with $CO_2$ and $C_4F_7N$. The adsorbents that are highly effective in the adsorption of HF, including NaF [17], $CaF_2$ [18], $MgF_2$, $Al(OH)_3$ [19], and the desiccants, including $Na_2SO_4$, $CaCl_2$ [20], $MgSO_4$ [21] were investigated for their adsorption performance with $CO_2$ and $C_4F_7N$, respectively. The results suggested that the 5A and F-03 molecular sieve materials are highly effective in adsorption of $CO_2$ or $C_4F_7N$ and are not suitable for using in $C_4F_7N/CO_2$, while some of the HF adsorbents and desiccants showed good compatibility with $C_4F_7N/CO_2$ and could be screened as potential candidates.

## 2. Materials and Methods

### 2.1. Chemical Reagents

The chemical reagents used in this study, including $Al(OH)_3$, $CaCl_2$, $MgSO_4$, $Na_2SO_4$, NaF, $MgF_2$, $CaF_2$ and the zeolite molecular sieve materials 3A, 4A were purchased from Sinopharm Co. Ltd. The adsorbents, 5A and F-03 zeolite molecular sieves, were offered by Shandong Taikai High Voltage Switchgear Co. Ltd. All of the chemicals with analytical grade or adsorbents were dried in an oven at 120 °C for 10 h to remove the moisture. Pure $CO_2$ (99.999%) used as a calibration gas was purchased from Xi'an Teda Cryogenic Equipment Co. Ltd.; and $C_4F_7N$ was purchased from a commercial market with a purity of 99%. The chemical composition of the zeolite molecular sieves are listed in Table 1.

**Table 1.** Chemical composition and pore size of molecular sieves.

| Molecular Sieve | Chemical Composition | Pore Size/nm |
|:---:|:---:|:---:|
| 3A | $Na_{6.6}K_{5.4}$-$[(AlO_2)_{12}(SiO_2)_{12}]$ | 0.3 |
| 4A | $Na_{12}\cdot[(AlO_2)_{12}\cdot(SiO_2)_{12}]$ | 0.4 |
| 5A | $Ca_6\cdot[(AlO_2)_{12}\cdot(SiO_2)_{12}]$ | 0.5 |
| F-03 | $Na_{12}\cdot[(AlO_2)_{12}\cdot(SiO_2)_{15}]$ | 1.0 |

### 2.2. Adsorption Characterization

To study the adsorption performance of the selected chemicals and adsorbents toward $C_4F_7N$ and $CO_2$, pulse adsorption tests were conducted in chemical adsorption equipment (Builder PCA-1200, Beijing Builder Co. Ltd., Beijing, China). The schematic and picture of the pulse adsorption test is

shown in Figure 1. When the pulse gas (tested gas) passed through the thermal conductivity detector (TCD), a pulse signal would show up, and the area of the signal peak is proportional to the amount of tested gas. Before testing the samples, the pulse adsorption procedure was run with an empty tube, and the obtained data were used as a blank control. To determine the adsorption performance of the samples, 0.05–0.20 g of each sample was filled in the sample tube, and the pulse adsorption procedures were conducted by turning a six-way valve to feed the calibration gas on a certain time interval. As shown in Figure 1A, for each test, in the first step, the quantitative loop was connected with the pulse gas line to fill with a fixed volume of pulse gas (0.30 mL), and in the six-way valve, gas passage 1 was connected with 6, while gas passage 4 was connected with 5. In the second step, the connection of the quantitative loop was switched to the sample tube, and in the six-way valve, gas passage 1 was connected with 2, and gas passage 4 was connected with 3. Then the carrier gas was purged and the pulse gas filled in the quantitative loop to pass through the sample and the TCD sequentially to record the pulse signal. Therefore, for each sample, 5–15 pulses were conducted depending on the adsorption performance of the testing sample, and the data obtained from the equipment were used to calibrate the adsorption capacity of the samples. For each sample, at least three tests were conducted, and the average data with less than 5% deviation were accepted.

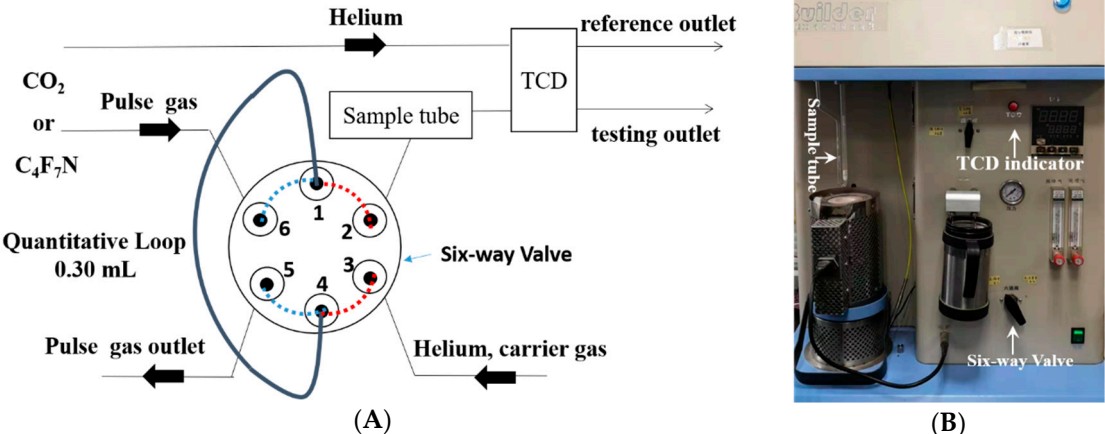

(**A**)  (**B**)

**Figure 1.** (**A**) Schematic and (**B**) picture of pulse adsorption test instrument (PCA-1200). TCD, thermal conductivity detector.

### 2.3. Data Analysis

The pulse signal data obtained were integrated to obtain the area of each peak. The area of each peak is proportional to the volume of calibration gas passed through the sample tube, and the difference of the area between the blank control was proportional to the amount of gas adsorbed by the samples. For each sample that adsorbed the target gas, several peaks with lower area than the control could be obtained, and the amount of gas adsorbed by the sample could be calculated according to the following equations

$$I_c = A_b / V \tag{1}$$

$$V_{ad} = \sum_{i=1}^{n} (A_b - A_i) / (m\, I_c) \tag{2}$$

where $A_b$ is the average area of peaks obtained with empty tubes for each tested calibrating gas in blank test; and $V$ represents volume of the quantitative loop in the six-way vale, which is 0.30 mL in this work. The item $I_c$ stands for the area of peaks for one milliliter calibrating gas; $A_i$ is the peaks with lower area compared with the control, when pulsing calibrating gas through sample in the tube. The item $n$ represents that the number of peaks showed lower integrated area than that of the control peaks and $m$ was the mass of the testing samples that filled the tube (in the unit of g). $V_{ad}$ represents the volume of calibrating gas adsorbed by the sample (mL/g).

## 3. Results and Discussion

### 3.1. Compatibility of Samples with $C_4F_7N$

Due to its relatively high boiling point, the content of $C_4F_7N$ used in the mixture gas is usually no more than 20% [22,23]. Therefore, the adsorbents or desiccants used to remove the moisture or acidic by-products from $C_4F_7N$ should not be able to adsorb $C_4F_7N$. Some of the moisture adsorbents, including 3A, 4A, 5A and F-03 molecular sieves, the desiccants including $Na_2SO_4$, $CaCl_2$, and the HF adsorbents, including $NaF$, $MgF_2$, $Al(OH)_3$ and $CaF_2$, were tested for their adsorption capacities on $C_4F_7N$ gas.

As shown in Figure 2, comparing with the pulse adsorption spectra using an empty tube as a control (Figure 2A), the 3A and 4A molecular sieves show slight adsorption capacity of $C_4F_7N$ (Figure 2B,C) with 0.39 and 1.44 mL/g, respectively as shown in Table 2. The 3A and 4A molecular sieves are usually used to dewater as they possess high surface areas and pore volumes [15], since the average pore sizes of 0.3 nm (for 3A molecular sieve) and 0.4 nm (for 4A molecular sieve) pore size are suitable to adsorb $H_2O$ molecules, however, it is calculated that the dynamic diameter for $C_4F_7N$ is around 0.7599 nm [12], which is significantly larger than the pore sizes of 3A and 4A molecular sieves. The surface area in micropores contributed most of the surface area, therefore, $C_4F_7N$ molecules are only able to adsorb on the surface of the 3A and 4A molecular sieves, which led to low adsorption capacity.

**Table 2.** Adsorption performance of the materials with $C_4F_7N$ based on the integrated area of pulse peaks.

| Items | Average Peak Area $A_b$/mV·s | $I_C$/mV·s/mL | Sample Mass/g | $V_{ad}$/mL/g |
|---|---|---|---|---|
| Blank control | 1059 ± 13 | 3530 | - | - |
| 3A | 1047 ± 3 | 3490 | 0.1437 | 0.39 |
| 4A | 1045 ± 8 | 3483 | 0.0463 | 1.44 |
| 5A | 741 | 2470 | 0.0658 | 22.82 |
| F-03 | 1055 ± 10 | 3516 | 0.1028 | 0.19 |
| $CaCl_2$ | 1060 ± 9 | 3533 | 0.1236 | - |
| $MgSO_4$ | 1055 ± 3 | 3516 | 0.1327 | 0.15 |
| $Na_2SO_4$ | 1056 ± 6 | 3520 | 0.1636 | 0.09 |
| $Al(OH)_3$ | 1056 ± 6 | 3520 | 0.1018 | 0.14 |
| NaF | 1057 ± 9 | 3523 | 0.1138 | 0.09 |
| $CaF_2$ | 1058 ± 5 | 3526 | 0.1042 | 0.05 |
| $MgF_2$ | 1049 ± 19 | 3496 | 0.1445 | 0.33 |
| $m(CaCl_2):m(Al(OH)_3) = 1:1$ | 1058 ± 10 | 3526 | 0.1426 | 0.04 |
| $m(CaCl_2):m(Al(OH)_3) = 2:1$ | 1057 ± 8 | 3523 | 0.1329 | 0.07 |

With 5A molecular sieve, although its average pore diameter is 0.5 nm, there are significant pores with sizes larger than 0.5 nm, besides, on the axial direction of this molecule, the diameter of the $CF_3$ group is smaller than 0.5 nm (0.4896 nm) [24], and therefore more surface area could be reachable for $C_4F_7N$ adsorption on 5A molecular sieve. As shown in Figure 2D, $C_4F_7N$ shows significant adsorption on 5A molecular sieve. The pulse adsorption peaks shown in Figure 2D are trailing, which suggests that the interaction between 5A molecular sieve and $C_4F_7N$ are strong. The adsorption capacity for $C_4F_7N$ is 22.82 mL/g calculated according to Equations (1) and (2). As for the commonly used adsorbent F-03, it also shows slight adsorption of $C_4F_7N$ as shown in Figure 2E, in which the intensity of the signal peaks is slightly lower than that of the blank control.

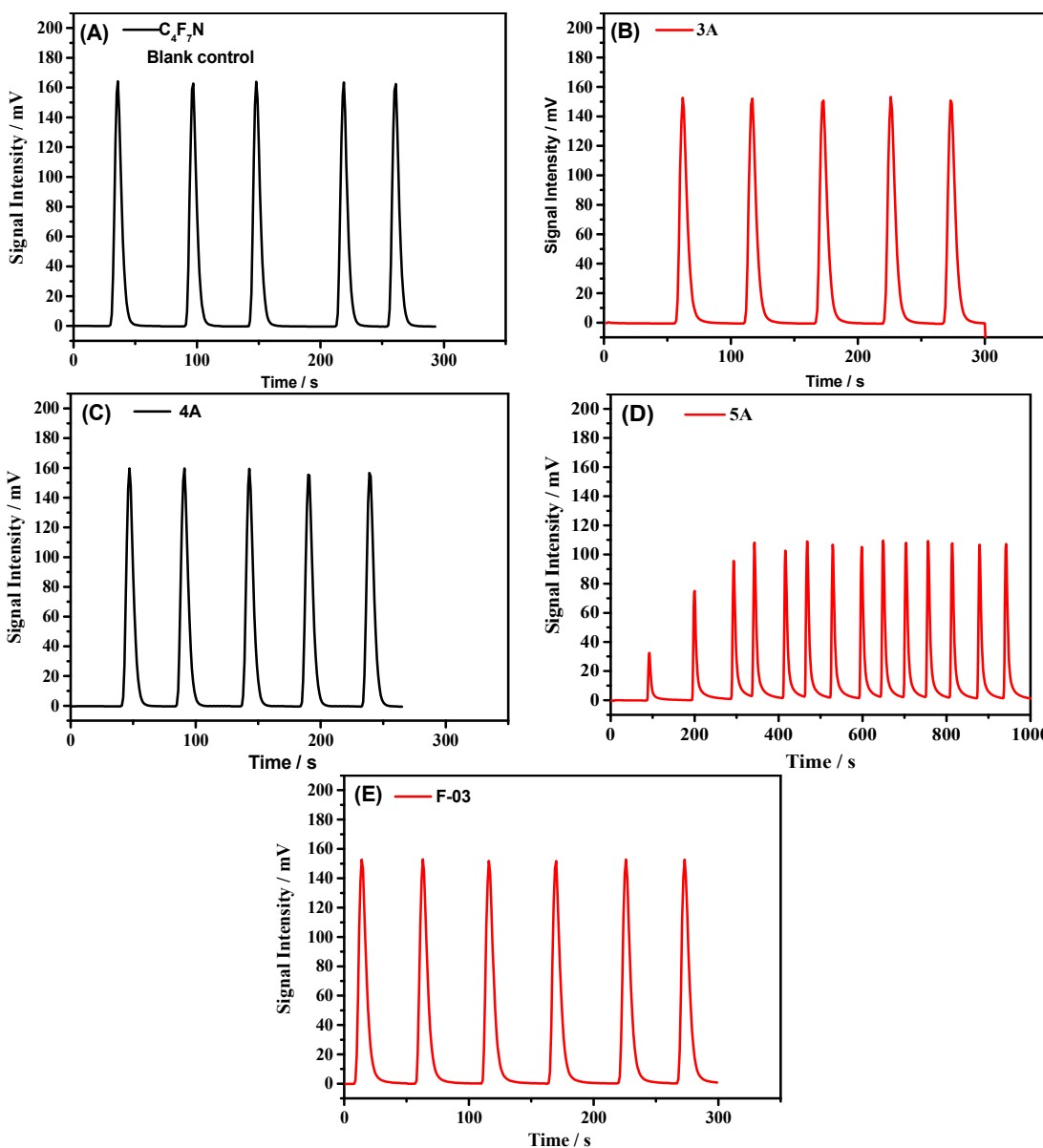

**Figure 2.** Adsorption performance of adsorbent materials with $C_4F_7N$ tested with pulse adsorption, (**A**) Blank control, (**B**) 3A, (**C**) 4A, (**D**) 5A molecular sieve, (**E**) F-03.

Since the 5A molecular sieves could adsorb $C_4F_7N$, it is not suitable to use these materials to eliminate the moisture from $C_4F_7N$ gas. One alternative strategy could be using the common desiccants, such as $CaCl_2$, $MgSO_4$, $Na_2SO_4$. These chemicals implement dewatering efficiently by forming crystal water. Since these chemicals possess low surface area, they should show negligible adsorption capacity of $C_4F_7N$. As shown in Figure 3, three desiccants, including $CaCl_2$, $MgSO_4$ and $Na_2SO_4$, show negligible adsorption with $C_4F_7N$. The adsorption capacity data listed in Table 2 also show that these chemicals do not intend to adsorb $C_4F_7N$. Therefore, these three desiccants could be used for dewatering of $C_4F_7N$ gas.

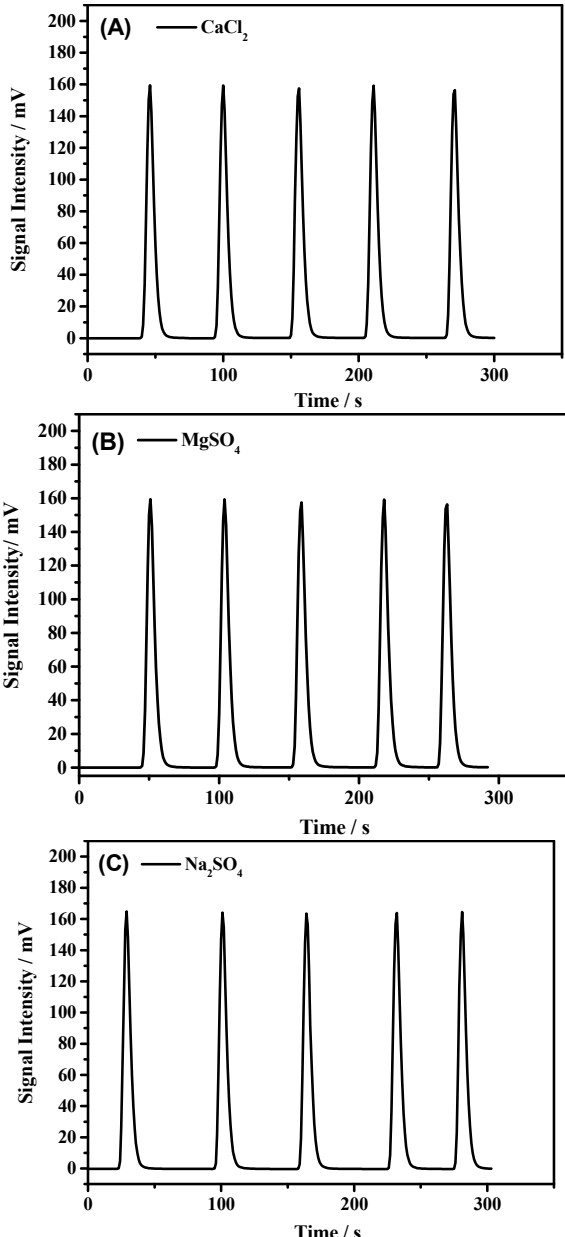

**Figure 3.** Adsorption performance of desiccants on $C_4F_7N$ tested with pulse adsorption, (**A**) $CaCl_2$, (**B**) $MgSO_4$, (**C**) $Na_2SO_4$.

Some fluorides are good HF adsorbents, including NaF [17,25], $MgF_2$ and $CaF_2$ [26]. Due to the reactivity with HF, $Al(OH)_3$ has also proved to be good HF remover [27]. These chemicals are potential HF removers that could be placed in the gas insulated electricity transfer equipment filled with $C_4F_7N$ gas. Therefore, the adsorption performances of these chemicals on $C_4F_7N$ are significant data. The ideal situation of negligible adsorption with this gas was expected to be observed. The pulse adsorption data are shown in Figure 4. As shown in these patterns, NaF, $CaF_2$ and $Al(OH)_3$ show negligible adsorption of $C_4F_7N$, while $MgF_2$ shows clear interaction with $C_4F_7N$. The adsorption capacity data listed in Table 2 also support the conclusion. These data suggest that NaF, $CaF_2$ and $Al(OH)_3$ are compatible with $C_4F_7N$ when used as a HF remover.

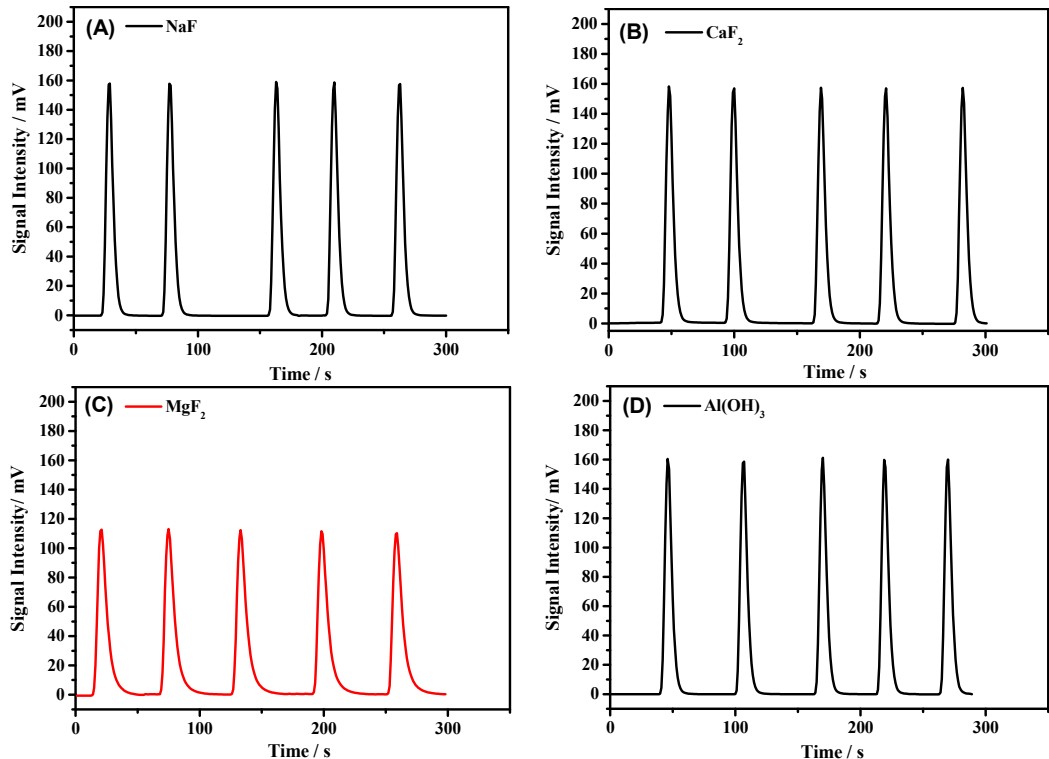

**Figure 4.** Adsorption performance of HF adsorbents on $C_4F_7N$, (**A**) NaF, (**B**) $CaF_2$, (**C**) $MgF_2$, (**D**) $Al(OH)_3$.

A mixture of desiccant ($CaCl_2$) and HF remover ($Al(OH)_3$) was also tested for its compatibility with $C_4F_7N$ gas. As shown in Figure 5, regardless if the mass ratio of desiccant to HF remover was 1 or 2, the mixture did not show clear adsorption performance on $C_4F_7N$. These data suggest that using a mixture of desiccant and HF remover to eliminate the moisture and HF could be a promising way to substitute the 5A or F-03 adsorbents.

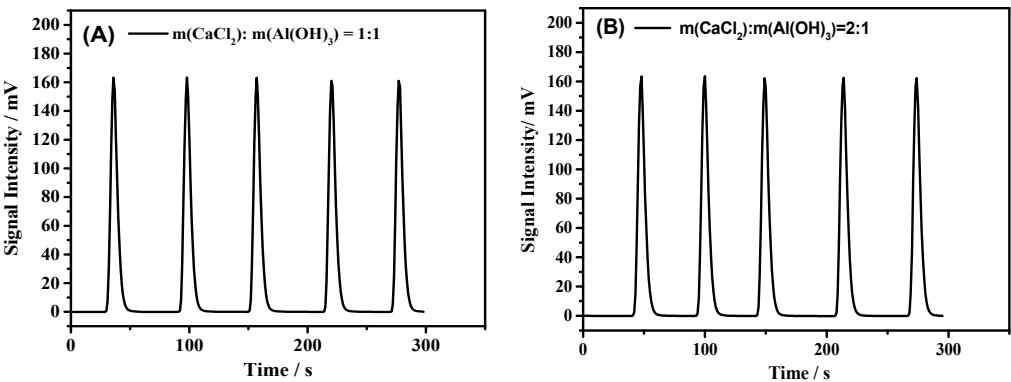

**Figure 5.** Adsorption performance of a mixture of desiccant and HF adsorbent, with a mass ratio of $CaCl_2$ to $Al(OH)_3$ equal to (**A**) 1:1, (**B**) 2:1.

## 3.2. Compatibility of Samples with $CO_2$

In $C_4F_7N/CO_2$, the ratio of $CO_2$ could be more than 90% (v/v), therefore, to remove moisture and HF, the compatibility of the adsorbents with $CO_2$ is significant. Both of $CO_2$ and HF are acidic gases, and the reactivity of the adsorbents with $CO_2$ may compromise the efficiency for HF removal. In this work, the compatibility of the above tested molecular sieves, including 3A, 4A, 5A, F-03, the desiccants,

such as $CaCl_2$, $MgSO_4$, $Na_2SO_4$, and HF remover, NaF, $MgF_2$, $CaF_2$ and $Al(OH)_3$ were tested with pulse adsorption procedures to determine their adsorption performance or interaction with $CO_2$.

As shown in Figure 6B,C, Figures 3A and 4A molecular sieves show slight adsorption of $CO_2$ compared with the blank control in Figure 6A, besides, the data listed in Table 3 show that the adsorption capacity is 0.8 mL/g and 3.13 mL/g, respectively. The 5A molecular sieve showed clear adsorption with $CO_2$, as shown in Figure 6D, and this result is consistent with the previous study [28]. The peak intensity is lower than the blank control and they are trailing clearly, which suggests the $CO_2$ is strongly interacting with the 5A molecular sieve. The adsorption capacity listed in Table 3 is 43.66 mL/g. It is well known that 5A molecular sieve has high adsorption capacity of $CO_2$ [11,28]. The F-03 adsorbents also show a high $CO_2$ adsorption capacity, which is 26.2 mL/g as listed in Table 3. Therefore, 5A molecular sieve and F-03 are not compatible with $C_4F_7N/CO_2$ insulating gas.

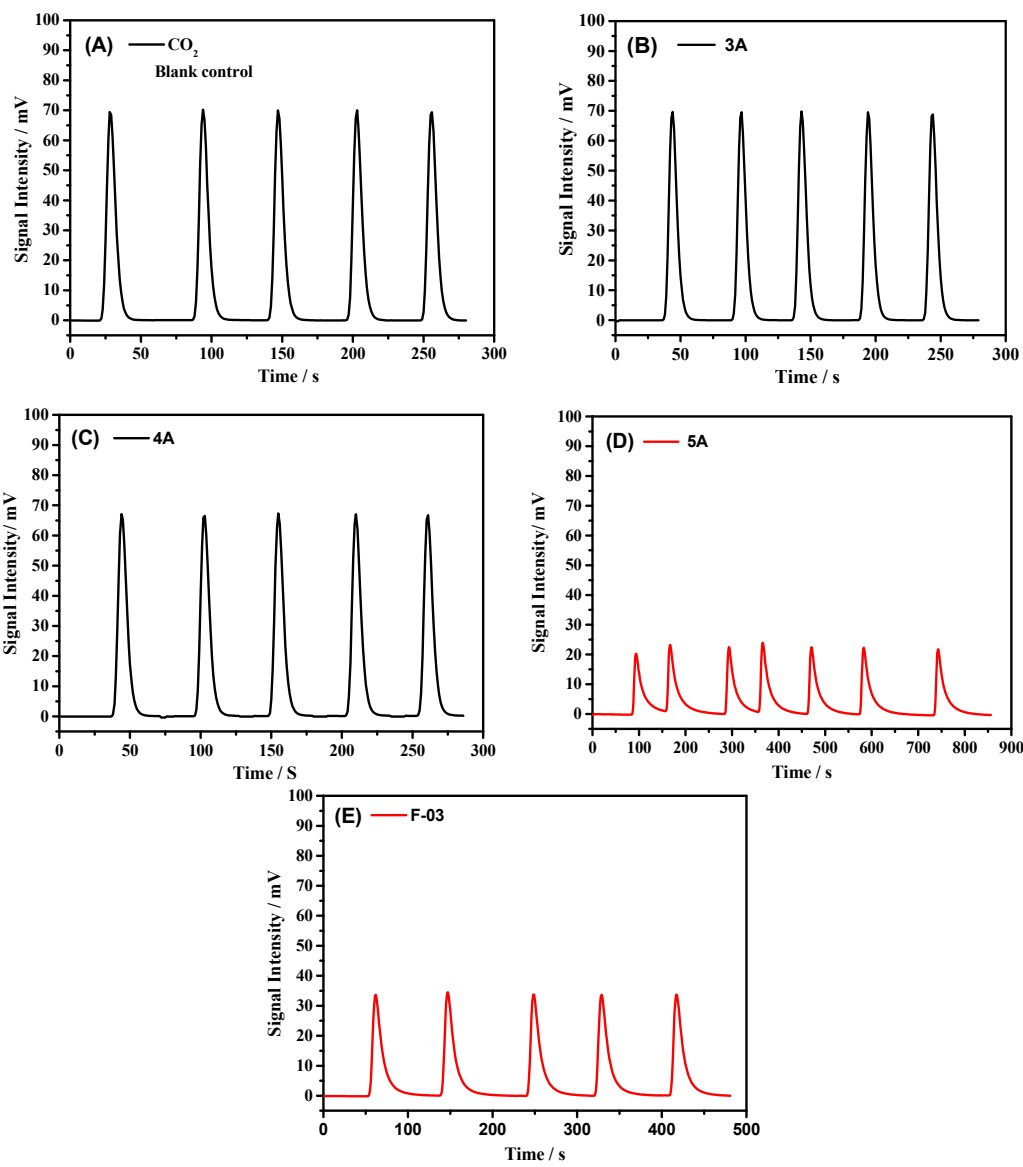

**Figure 6.** Adsorption performance of adsorbent materials with $CO_2$ tested with pulse adsorption, (**A**) Blank control, (**B**) 3A, (**C**) 4A, (**D**) 5A molecular sieve, (**E**) F-03.

**Table 3.** Adsorption performance of the materials on $CO_2$ based on the integrated area of pulse peaks.

| Items | Average Peak Area $A_b$/mV·s | $I_C$/mV·s/mL | *Sample Mass*/g | $V_{ad}$/mL/g |
|---|---|---|---|---|
| Blank control | 523 ± 4 | 1743 | - | - |
| 3A | 516 ± 4 | 1720 | 0.0824 | 0.80 |
| 4A | 499 ± 4 | 1663 | 0.073 | 3.13 |
| 5A * | 430 | 1433 | 0.0407 | 43.66 |
| F-03 * | 427 | 1423 | 0.07 | 26.20 |
| $CaCl_2$ | 524 ± 6 | 1747 | 0.0506 | 0 |
| $MgSO_4$ | 525 ± 3 | 1750 | 0.0682 | 0 |
| $Na_2SO_4$ | 528 ± 5 | 1760 | 0.1317 | 0 |
| $Al(OH)_3$ | 522 ± 2 | 1740 | 0.1285 | 0.07 |
| NaF | 520 ± 4 | 1733 | 0.1328 | 0.21 |
| $CaF_2$ | 518 ± 7 | 1727 | 0.1233 | 0.38 |
| $MgF_2$ | 518 ± 6 | 1727 | 0.1428 | 0.33 |
| $m(CaCl_2):m(Al(OH)_3) = 1:1$ | 519 ± 3 | 1730 | 0.1235 | 0.30 |
| $m(CaCl_2):m(Al(OH)_3) = 2:1$ | 512 ± 3 | 1707 | 0.1326 | 0.79 |

* 5A molecular sieves and F-03 were tested for 10 cycles, and the others were tested for five cycles.

Similar with the results tested in $C_4F_7N$, the three desiccants did not show clear adsorption with $CO_2$, as shown in Figure 7, and the data listed in Table 3. The data also suggest the three chemicals would not react with $CO_2$. Since no clear adsorption with $C_4F_7N$ was observed, they could be used for removing the moisture in the insulating gas $C_4F_7N/CO_2$.

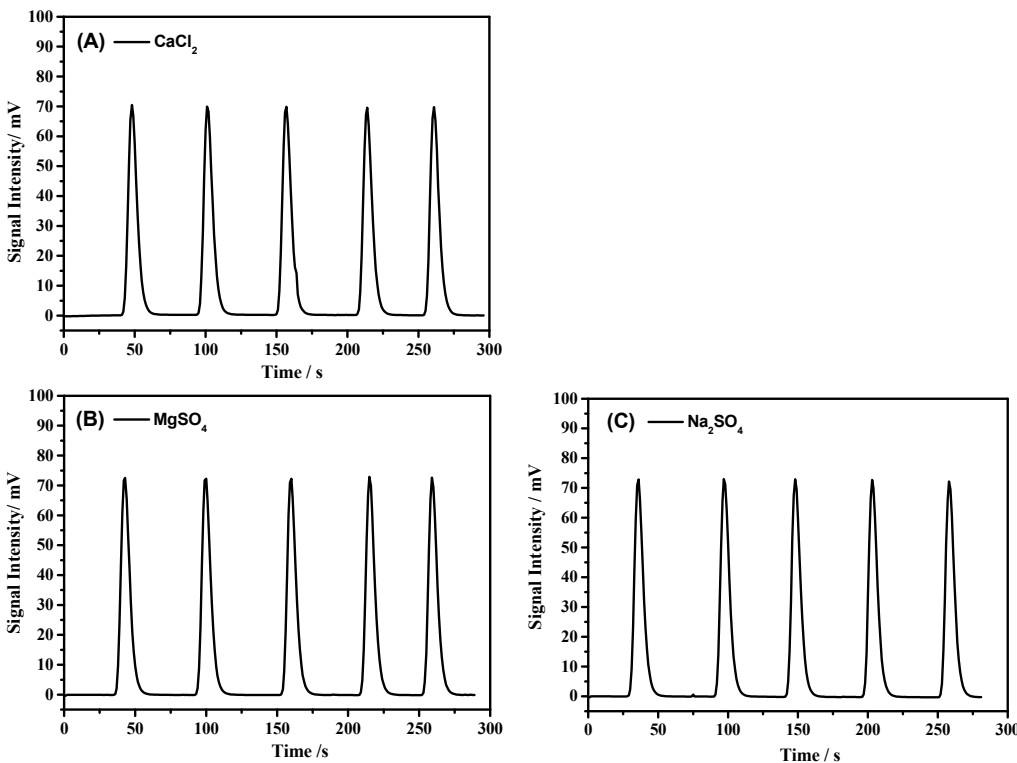

**Figure 7.** Adsorption performance of desiccants on $CO_2$ tested with pulse adsorption, (**A**) $CaCl_2$, (**B**) $MgSO_4$, (**C**) $Na_2SO_4$.

All of the four HF removers are alkaline chemicals, one would suspect that these chemicals may react with $CO_2$. The pulse adsorption data presented in Figure 8 suggest that the four chemicals show negligible adsorption of $CO_2$, and the adsorption capacity data listed in Table 3 are all below 0.5 mL/g. These data suggest that $CO_2$ would not react with the four HF removers. The p$K_a$ of HF is 3.18, and

the p$K_{a1}$ of $H_2CO_3$ is 6.38, therefore, the fluoride salts are stable in $CO_2$ gas. $Al(OH)_3$ is a weak alkali, and it is also stable in $CO_2$ gas.

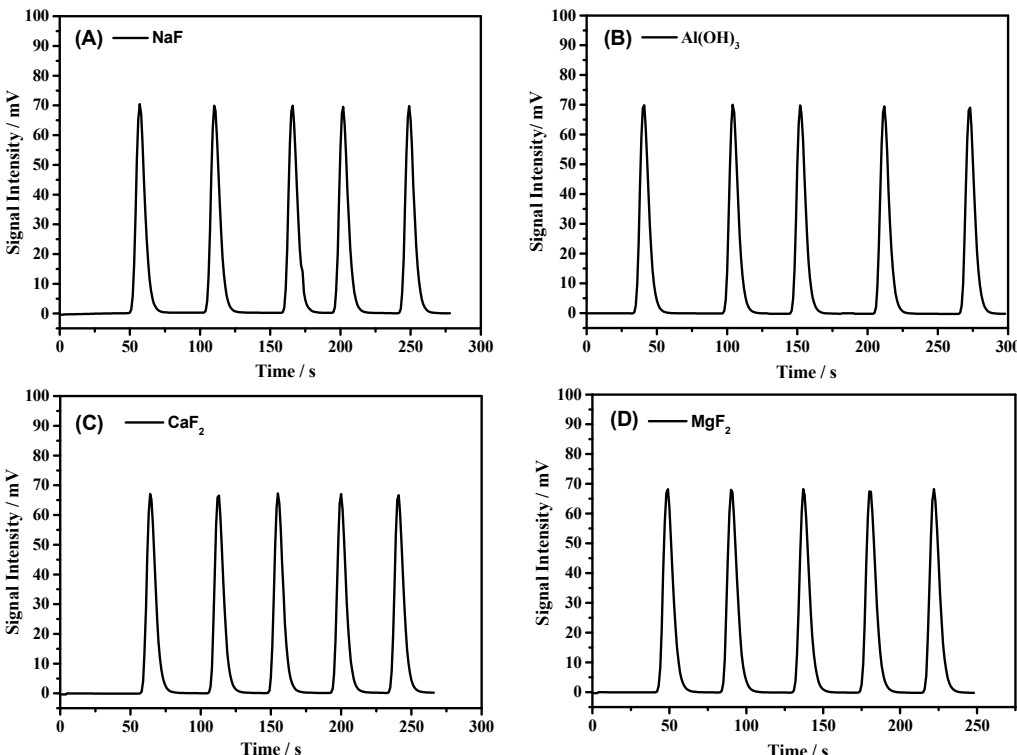

**Figure 8.** Adsorption performance of HF adsorbents on $CO_2$, (**A**) NaF, (**B**) $CaF_2$, (**C**) $MgF_2$, (**D**) $Al(OH)_3$.

Since both the desiccants and HF remover studied in this work did not show clear reaction or adsorption with $CO_2$, logically, the mixture of a desiccant and HF remover should also not adsorb or react with $CO_2$. The data shown in Figure 9 and Table 3 prove that the mixture of $CaCl_2$ and $Al(OH)_3$ are compatible in $CO_2$, which is the same result as tested in $C_4F_7N$. Therefore, the mixture of desiccants with HF remover could be used in $C_4F_7N/CO_2$.

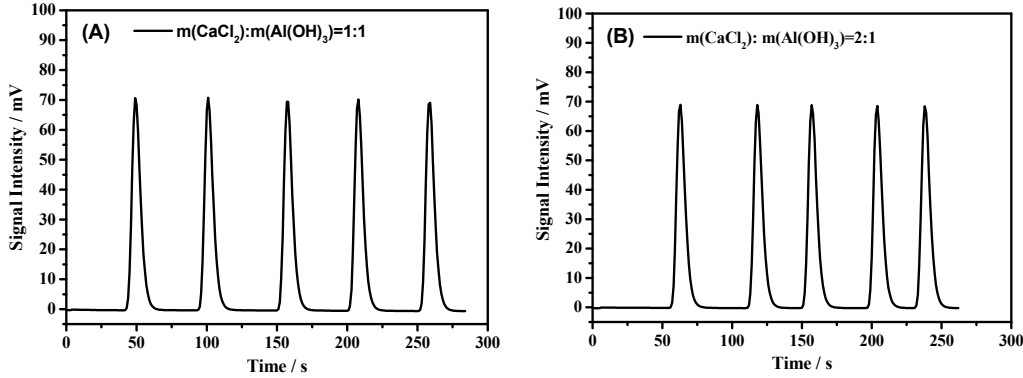

**Figure 9.** Adsorption performance of a mixture of desiccant and HF adsorbent on $CO_2$, with a mass ratio of $CaCl_2$ to $Al(OH)_3$ equal to (**A**) 1:1, (**B**) 2:1.

## 4. Conclusions

The pulse adsorption tests suggested that the commonly used adsorbents 5A and F-03 molecular sieves could not be used in $C_4F_7N/CO_2$, due to the severe adsorption of the mixed gas on these molecular sieves. The 3A and 4A molecular sieves adsorb $C_4F_7N$ and $CO_2$ slightly, and might be

used as adsorbents for $C_4F_7N/CO_2$. Desiccants, including $Na_2SO_4$, $CaCl_2$ and $MgSO_4$ show negligible adsorption with $C_4F_7N$ and $CO_2$. Some HF removers, such as $NaF$, $CaF_2$, $Al(OH)_3$ also show negligible adsorption with the two gases, and could be compatible with them sealed in related gas insulated electricity transfer equipment. Using a mixture of desiccant and HF remover could be a good strategy to remove the moisture and HF produced in the $C_4F_7N/CO_2$ insulated equipment.

**Author Contributions:** Investigation, Q.H.; funding acqusition Y.W.; methodology, J.L.; Investigation Y.Z.; Investigation L.Z.

**Funding:** This research was funded by the project 'Study on Physical, Chemical and Insulation Properties, and Engineering Demostration of Environmental Insulating gas (I)-Project 3-Applied Feasibility Study of New Insulating Gas in Guangzhou Power Grid', numbered as GZJKJXM20170330.

**Conflicts of Interest:** The authors declare no conflict of interest.

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
