# Peer review of "Study on the Compatibility of Gas Adsorbents Used in a New Insulating Gas Mixture C4F7N/CO2"

_processes, doi:10.3390/pr7100698_

Round 1
Reviewer 1 Report
In this work, a strategy to use a mixture of desiccants (including CaCl2, Na2SO4, MgSO4) and HF adsorbents (including NaF, CaF2, Al(OH)3) in the equipment, filled with new C4F7N/CO2 insulating gas, instead of 5 A and F-03 molecular sieves used for SF6 insulating gas is suggested on a base of pulse adsorption tests.
In general, this manuscript can be considered for publication, although pulse adsorption test is the only kind of performed experiment that is evident drawback of the manuscript. If authors could add some electrical tests at least for the best case, it would improve the manuscript.
Another drawback is poor clearance and accuracy of the manuscript.
Thus, in abstract, many symbols have to be subscripted but they are not, while text can be more clear and short.
In introduction, no kind of application for insulating gases is presented. Moreover, C4F7N and SF6 insulating gases are compared between themselves. However, in the middle of the manuscript it appears that for practical use only 10% C4F7N and 90% CO2 gas mixture should be considered, while no information for CO2 is given in the introduction. As a result, an importance of this study is unclear without additional information on the questions above. Furthermore, in contrast to desiccants and adsorbents with clear chemical composition, the materials of the molecular sieves are not indicated. The difference between them is also not shown until the middle of the manuscript, although for F-03 is it not shown at all that is not acceptable and requires more information on the sieves in the introduction. Finally, authors should use the same designation for the sieves as either 3A, 4A, 5A and F-03 or 3 A, 4 A, 5 A and FO-3 in introduction and across the manuscript text and figures.
Concerning smaller inaccuracies:
at line 36 it should be corrected to “decomposed” at line 78 to “calibration” Point is needed at the end of line 100 but not at line 91, while line 94 should start with noncapital letter without indentation at line 109 to “comparing” at lines 110 and 126 to ”slight” at line 115 to “significantly” at lines 137 and 219 to “shown” at lines 147-148 to “situation of negligible adsorption” at line 205 “show in Figure 8” to “presented at Figure 8” at lines 218-219 to “adsorb of react with” at line 230 “;” should be changed for “.” in Fig. 1 to “Quantitative loop”, while TCD has to be decoded.
Finally, figure labels (A), (B), etc. seem to hide some data in Figures 2-5 and 7. Therefore, these figures have to be modified to avoid that.
Thus, current manuscript can be published in Processes when its subject and importance is clarified, particularly, in the introduction part, additional experiments are considered and inaccuracies are corrected.
Author Response
Dear Reviewer,
We appreciate your time for reviewing this manuscript. We have revised the manuscript carefully according to your comments, and the revised words/sentences are marked with red color. The detail responses are listed below.
Best Regards!
Jing Liu
Reviewer 1#
Comment 1: In general, this manuscript can be considered for publication, although pulse adsorption test is the only kind of performed experiment that is evident drawback of the manuscript. If authors could add some electrical tests at least for the best case, it would improve the manuscript.
Response 1:We appreciate your advice. Currently, the purpose of this study is to study the compatibilty of the common adsorbants on the new insulating gas. The adsorption of gasous molecules is dominated by the surficail characteristics and porous structure of adsorbants. According to the results of this work, we can figure out whether the common adsorbants are acceptable, or we could use alternative strategy by mixing desiccant with HF adsorbants to eliminate moisture and HF simultaneous. Furthermore, according to the results of this work, we will also try to tune the porous structure of the common adsorbants to make it compatible with the new insulating gas C4F7N/CO2. Therefore, based on these reasons, only the pulse adsorption test is indispensable in this work.
Comment 2: Another drawback is poor clearance and accuracy of the manuscript. Thus, in abstract, many symbols have to be subscripted but they are not, while text can be more clear and short.
Response 2: We appreciate your comment, we have revised all of the mistakes on subscript of molecular formula.
Comment 3: In introduction, no kind of application for insulating gases is presented. Moreover, C4F7N and SF6 insulating gases are compared between themselves.
Response 3: We appreciate your comment. In the introduction section, the application background for insulating gases are described as below:
Currently, SF6 is the most widely used insulating gas in electrical insulation equipments, such as gas insulated switchgear and gas insulated line, however, due to its environmental issues, a new environmental friendly insulating gas is urgently needed.
Since SF6 is the most widely used insulating gas, the characteristics of C4F7N is compared with that of SF6 to show its potential as an alternative insulating gas.
Comment 4: However, in the middle of the manuscript it appears that for practical use only 10% C4F7N and 90% CO2 gas mixture should be considered, while no information for CO2 is given in the introduction. As a result, an importance of this study is unclear without additional information on the questions above.
Response 4: We appreciate your comment. As we know, CO2 is a commonly used gas, and it is used as a buffer gas in the mixture of C4F7N/CO2 due to the high boiling point of C4F7N. With the addition of CO2, the pressure of C4F7N/CO2 could be at some point higher than that of the atmosphere. While in the mixture of C4F7N/CO2, C4F7N is the primary component for the insulating performance of the mixture gas. Therefore, the compatibility of adsorbants with C4F7N is essential for maintaining its insulating performance.
Comment 5: Furthermore, in contrast to desiccants and adsorbents with clear chemical composition, the materials of the molecular sieves are not indicated. The difference between them is also not shown until the middle of the manuscript, although for F-03 is it not shown at all that is not acceptable and requires more information on the sieves in the introduction.
Response 5: We appreciate your advice. The chemical composition of the molecular sieves are listed in table 1.
Comment 6: Finally, authors should use the same designation for the sieves as either 3A, 4A, 5A and F-03 or 3 A, 4 A, 5 A and FO-3 in introduction and across the manuscript text and figures.
Response 6: We appreciate for your advice. We have corrected the mistakes, and use 3A, 4A, 5A, F-03 to designate the sieves.
Comment 7: Concerning smaller inaccuracies: at line 36 it should be corrected to “decomposed” at line 78 to “calibration” Point is needed at the end of line 100 but not at line 91, while line 94 should start with noncapital letter without indentation at line 109 to “comparing” at lines 110 and 126 to ”slight” at line 115 to “significantly” at lines 137 and 219 to “shown” at lines 147-148 to “situation of negligible adsorption” at line 205 “show in Figure 8” to “presented at Figure 8” at lines 218-219 to “adsorb of react with” at line 230 “;” should be changed for “.” in Fig. 1 to “Quantitative loop”, while TCD has to be decoded.
Response 7: We appreciate your advice. We revised the manuscript according to your suggestions.
Comment 8: Finally, figure labels (A), (B), etc. seem to hide some data in Figures 2-5 and 7. Therefore, these figures have to be modified to avoid that. Thus, current manuscript can be published in Processes when its subject and importance is clarified, particularly, in the introduction part, additional experiments are considered and inaccuracies are corrected.
Response 8: We appreciate your comments. The figure labels have been revised properly to avoid hiding some data in these figures.

Reviewer 2 Report
Dear authors,
The manuscript is focused on the comparison of gas absorbants compatibility with the insulating gas C4F7N/CO2. Please update the manuscript with the comments suggested below to improve the quality of the manuscript.
Please provide the chemical composition of 5 A molecular sieve and F-03 in the abstract to help the reader understand their exact chemical nature. Please include the quantification of data in the abstract. For example: Please include the actual increase of 5A molecular sieve adsorption property C4F7N (22.82 mL/g) and CO2 (43.86 mL/g). Please update the chemical formulae of all the chemical with the proper subscript font in the whole paper. For example CaCl2, Na2SO4 and MgSO4 to CaCl2, Na2SO4 and MgSO4. In Introduction (line 56): Please details of the equipment instead of just mentioning as Chinese electrical equipment, so that readers can have more details understanding. Please include any specific reason to choose the range of sieve materials chooses to test from 3A. 4A and 5A. Please include the references from the literature and peer publications for considering the use of specific sieve materials considered for this study. Please update the quality of the schematic image of pulse adsorption test instrument and include more details of the test process in the text. Also, please include the image of the actual instrument used to have a realistic understanding of the equipment. In the results and discussion session please include the references of the supporting literature and published materials to draw the conclusions. In the current version, there are not enough references included to support the argument along with the experiments. Please include the scientific reasoning to your observations in the results and discussion session. The current version of the manuscript is only explaining and detailing the results instead of providing scientific reasoning. In conclusion, also, the results need to be quantified properly to make a proper evaluation of the manuscript. For example: when mentioned about the increase or decrease of the adsorption, the percentage increase needs to be mentioned. Please improve English writing in the manuscript.
Author Response
Reviewer 2#
The manuscript is focused on the comparison of gas absorbants compatibility with the insulating gas C4F7N/CO2. Please update the manuscript with the comments suggested below to improve the quality of the manuscript.
Comment 1: Please provide the chemical composition of 5 A molecular sieve and F-03 in the abstract to help the reader understand their exact chemical nature. Please include the quantification of data in the abstract. For example: Please include the actual increase of 5A molecular sieve adsorption property C4F7N (22.82 mL/g) and CO2 (43.86 mL/g).
Response 1: We appreciate your comments. We provide the chemical composition of 5A and F-03 molecular sieve in section 2.1, table 1. The quantification data are included in the abstract.
Comment 2: Please update the chemical formulae of all the chemical with the proper subscript font in the whole paper. For example CaCl2, Na2SO4 and MgSO4 to CaCl2, Na2SO4 and MgSO4.
Response 2: Thank you to your comment, and we have corrected the mistakes of improper subscript font.
Comment 3: In Introduction (line 56): Please details of the equipment instead of just mentioning as Chinese electrical equipment, so that readers can have more details understanding.
Response 3: We provide the equipments that could use the insulating gas in the introduction section, such as gas insulated switchgear and gas insulated line.
Comment 4: Please include any specific reason to choose the range of sieve materials chooses to test from 3A. 4A and 5A.
Response 4: The reason for using 3A, 4A and 5A molecular sieves are included in the introduction section. The reason is that these materials are commercial desiccants and adsorbants. Some references are also supplemented.
Comment 5: Please include the references from the literature and peer publications for considering the use of specific sieve materials considered for this study.
Response 5: The refereces from the literature for considering the use of the specific sieve materials are supplemented.
Comment 6: Please update the quality of the schematic image of pulse adsorption test instrument and include more details of the test process in the text. Also, please include the image of the actual instrument used to have a realistic understanding of the equipment.
Response 6: We appreciate your comments. We have revised the schematic image of pulse adsorption test instrument to make it clear to show the gas flow lines between pulse gas filling and sampling, and we also provide picture of the acutual instrument in figure 1.
Comment 7: In the results and discussion session please include the references of the supporting literature and published materials to draw the conclusions.
Response 7:We appreciate your comments. Several references of the supporting literature and published materials are supplemented to draw the conclusions.
Comment 8: In the current version, there are not enough references included to support the argument along with the experiments.
Response 8:We appreciate your advice, and some references are supplemented in the discussion section to support the argument along with the experiments.
Comment 9: Please include the scientific reasoning to your observations in the results and discussion session.
Response 9:We appreciate your advice, and the scientific reasoning to our observations in the results and discussion session are supplemented. As shown in the tables and figures, 5A and F-03 would adsorb C4F7N or CO2, which are the components of the insulating gas mixture C4F7N/CO2, and could decrease the insulating performance of the insulating gas. Therefore, these two molecular sieves are not compatible with C4F7N/CO2. While some desiccants, such as Na2SO4, CaCl2 and MgSO4, and some HF removers, such as NaF, Al(OH)3 and CaF2, could not adsorb the components of C4F7N/CO2, and therefore, these chemicals are potential candidates using as moisture and HF removers in C4F7N/CO2 filled electricity transfer equipments.
Comment 10: The current version of the manuscript is only explaining and detailing the results instead of providing scientific reasoning. In conclusion, also, the results need to be quantified properly to make a proper evaluation of the manuscript. For example: when mentioned about the increase or decrease of the adsorption, the percentage increase needs to be mentioned. Please improve English writing in the manuscript.
Response 10:We appreciate your comments. We have supplement the scientific reasoning as we respond in response 9. For the adsorption test, the purpose of this study was to evaluate the compatible of the selected adsorbants and chemicals with the insulating gas C4F7N/CO2. All of the adsorption data are independent, and were not comparing with other data, therefore, no decrease or increase percentage would be mentioned. The English of this manuscript has been revised.

Round 2
Reviewer 1 Report
In the revised manuscript, most of the comments of the previous report are addressed, although not all of them completely.
Besides that the authors did not present any additional experiment like electrical tests for the best case under study, they state that in 10% C4F7N and 90% CO2 gas mixture, C4F7N is the primary component for the insulating performance. If that so, the authors have to explain clearly why and support the explanation by references.
Then, if the sieve materials are of zeolite nature, that should be indicated in the manuscript text as well.
Finally, there are still 5 A instead of 5A sieve name in the lines 24, 140 and Figure 2D, while 3 A is in Figure 6B. Moreover, figure labels (A), (B), etc. are better to place right in the figure files, but not in Word file of the manuscript. Furthermore, the figures have to be arranged to avoid jumps from one page to another.
In addititon, in tables 2 and 3, “mV·s·mL-1” have to be unified with other units as “mV·s/mL”
Thus, current manuscript can be published in Processes after the minor corrections.
Author Response
Dear Reviewer:
We appreciate your comments and suggestions, and we have revised the manuscript according to your advices. The responses are listed below point by point.
Best Regards!
Jing Liu
Comment 1: In the revised manuscript, most of the comments of the previous report are addressed, although not all of them completely. Besides that the authors did not present any additional experiment like electrical tests for the best case under study, they state that in 10% C4F7N and 90% CO2 gas mixture, C4F7N is the primary component for the insulating performance. If that so, the authors have to explain clearly why and support the explanation by references.
Response 2: We appreciate your comments. Two references ([22] Zhang XX, Zhang QC, Zhang J, Li Y, Xiao S, Zhuo R, Tang J. Experimental study on power frequency breakdown characteristics of C4F7N/CO2 gas mixture under quasi-homogeneous electric field. IEEE Access. 2019;7:19100-8.[23] Li Z, Ding W, Liu Y, Li Y, Zheng Z, Liu W, et al. Surface flashover characteristics of epoxy insulator in C4F7N/CO2 mixtures in a uniform field under AC voltage. IEEE Trans Dielect Elect Insul. 2019;26:1065-72.) are supplemented in Line 121. And in these two references, the concentration of C4F7N was no more than 20% due to its relatively high boiling point.
Comment 2: Then, if the sieve materials are of zeolite nature, that should be indicated in the manuscript text as well.
Response 2: We appreciate your comments. The zeolite nature of molecular sieve materials are indicated in line 60,62, 72 and 73.
Comment 3: Finally, there are still 5 A instead of 5A sieve name in the lines 24, 140 and Figure 2D, while 3 A is in Figure 6B. Moreover, figure labels (A), (B), etc. are better to place right in the figure files, but not in Word file of the manuscript. Furthermore, the figures have to be arranged to avoid jumps from one page to another.
Response 3: We appreciate your comments. We have corrected the mistakes on the lines 24, 140 and Figure 2D, 6B; we also revised the figure labels according to your advices.
Comment 4: In addititon, in tables 2 and 3, “mV·s·mL-1” have to be unified with other units as “mV·s/mL”. Thus, current manuscript can be published in Processes after the minor corrections.
Response 4: We appreciate your suggestion, and we revised the tables according to your advice.
Reviewer 2 Report
Dear Authors,
Thank you for your updated version of the manuscript as per the comments mentioned in the review report.
Author Response
Dear Reviewer,
We appreciate your comments and suggestions on our work. According to your advices, we have revised the manuscript thoroughly, and the revised sentences or words are marked in red color.
Best Regards!
Jing Liu